# Identification of a peptide recognizing cerebrovascular changes in mouse models of Alzheimer's disease

Aman P. Mann[1], Pablo Scodeller[1,2], Sazid Hussain[1,3], Gary B. Braun[1], Tarmo Mölder[2], Kadri Toome[2], Rajesh Ambasudhan[4], Tambet Teesalu[2], Stuart A. Lipton[4,5] & Erkki Ruoslahti[1,6]

Cerebrovascular changes occur in Alzheimer's disease (AD). Using in vivo phage display, we searched for molecular markers of the neurovascular unit, including endothelial cells and astrocytes, in mouse models of AD. We identified a cyclic peptide, CDAGRKQKC (DAG), that accumulates in the hippocampus of hAPP-J20 mice at different ages. Intravenously injected DAG peptide homes to neurovascular unit endothelial cells and to reactive astrocytes in mouse models of AD. We identified connective tissue growth factor (CTGF), a matricellular protein that is highly expressed in the brain of individuals with AD and in mouse models, as the target of the DAG peptide. We also showed that exogenously delivered DAG homes to the brain in mouse models of glioblastoma, traumatic brain injury, and Parkinson's disease. DAG may potentially be used as a tool to enhance delivery of therapeutics and imaging agents to sites of vascular changes and astrogliosis in diseases associated with neuroinflammation.

[1] Cancer Research Center, Sanford Burnham Prebys Medical Discovery Institute, La Jolla, CA 92037, USA. [2] Laboratory of Cancer Biology, Institute of Biomedicine and Translational Medicine, University of Tartu, Tartu 50411, Estonia. [3] AivoCode Inc., La Jolla, CA 92037, USA. [4] Neurodegenerative Disease Center, Scintillon Institute, San Diego, CA 92121, USA. [5] Department of Neurosciences, University of California, San Diego, School of Medicine, La Jolla, CA 92093, USA. [6] Center for Nanomedicine and Department of Cell, Molecular and Developmental Biology, University of California, Santa Barbara, Santa Barbara, CA 93106, USA. Aman P. Mann and Pablo Scodeller contributed equally to this work. Correspondence and requests for materials should be addressed to E.R. (email: ruoslahti@sbpdiscovery.org)

Alzheimer's disease (AD) is the most common progressive neurodegenerative disorder associated with aging. The prevalent hypothesis regarding AD pathogenesis is based on the premise that deposition of amyloid-β peptide (Aβ), followed by hyperphosphorylated tau deposits, plays a key role in neurodegeneration and brain atrophy in AD. However, failure to demonstrate clinical efficacy by targeting Aβ[1] in all but one recent clinical study[2], dictates the necessity of identifying additional cellular pathways, processes, and molecules involved in AD pathogenesis for diagnosis and treatment to improve clinical outcomes.

Cerebrovascular changes in AD animal models and human patients have been reported[3, 4]. These morphological and functional changes can contribute to neuronal dysfunction and neurodegeneration[5]. For instance, degeneration of blood-brain barrier (BBB)-associated pericytes can lead to impairment of the BBB[6, 7, 8], resulting in serum buildup of proteins and edema[9], increase in reactive oxygen species[10], and neuronal injury[11]. However, this disruption in microvascular integrity also provides a therapeutic opportunity to access the brain extravascular space from the systemic circulation for delivering therapeutics and imaging agents to the AD brain.

In vivo peptide phage display can be used for unbiased probing of tissues in situ for specific molecular signatures, particularly in the vasculature[12]. We have successfully utilized this technique to discover homing peptides specific for different pathologies including tumors, atherosclerotic plaques, wounds and severe brain injury[13–15]. These homing peptides have been used to selectively target drugs, antibodies, and nanoparticles into pathological sites, and some peptides have advanced into clinical trials[14]. Given the changes reported in the neurovascular unit in AD, we conducted in vivo phage display to identify peptides that specifically recognize molecular changes associated with AD pathogenesis, and enable targeting of such sites from systemic administration.

## Results

**In vivo phage screening in a mouse AD model.** To identify peptides specific for AD brain, we performed in vivo phage display using a T7 phage library that displays 9-amino acid cyclic

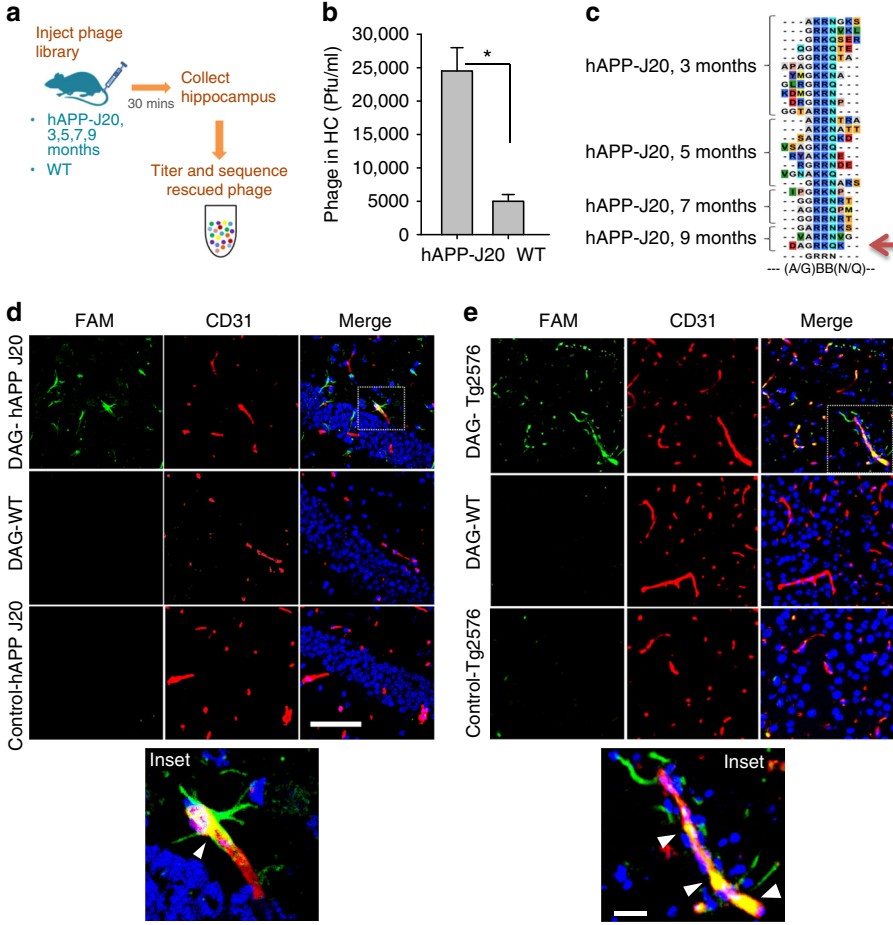

**Fig. 1** Identification of DAG peptide by phage screening in transgenic mouse model of AD. Schematic representation of phage screening done in transgenic hAPP-J20 tg mice **a**. A CX7C library ($10^9$ pfu) was injected intravenously in hAPP-J20 mice and wild-type (WT) littermate controls of different ages. After 30 min of circulation and perfusion to remove unbound phage, the hippocampus was excised and phages were recovered and quantified **b**. The phage DNA was subjected to high-throughput sequencing, and the sequences present in the hAPP-J20 brains were compared to sequences from WT brains **c**. Analysis of the hAPP-J20-specific sequences revealed a consensus motif, (A/G)BB(N/Q) (where B is basic). A cyclic 9-amino acid consensus peptide (*red arrow*) containing this motif (sequence CDAGRKQKC; abbreviated "DAG") was chemically synthesized. The DAG peptide was tested in the hAPP-J20 mice **d** and in another AD model, Tg2576 mice **e**. FAM-labeled DAG peptide was intravenously injected into 9-month old hAPP-J20 and 18-month old Tg2576 mice and allowed to circulate for 30 min. The mice were perfused, the brains were fixed, sectioned and stained with anti-FAM (green channel) to detect DAG and with anti-CD31 (red channel) for blood vessels. Homing of DAG was mainly seen in the hippocampus of the J20 mice and in the cortex of the Tg2576 mice, partially co-localizing with blood vessels. $n = 5$ mice brains were examined from each model. Representative images are shown. *Scale bars*, 50 μm (panels **d** and **e**), 10 μm (*insets* in **d** and **e**). *$P < 0.05$

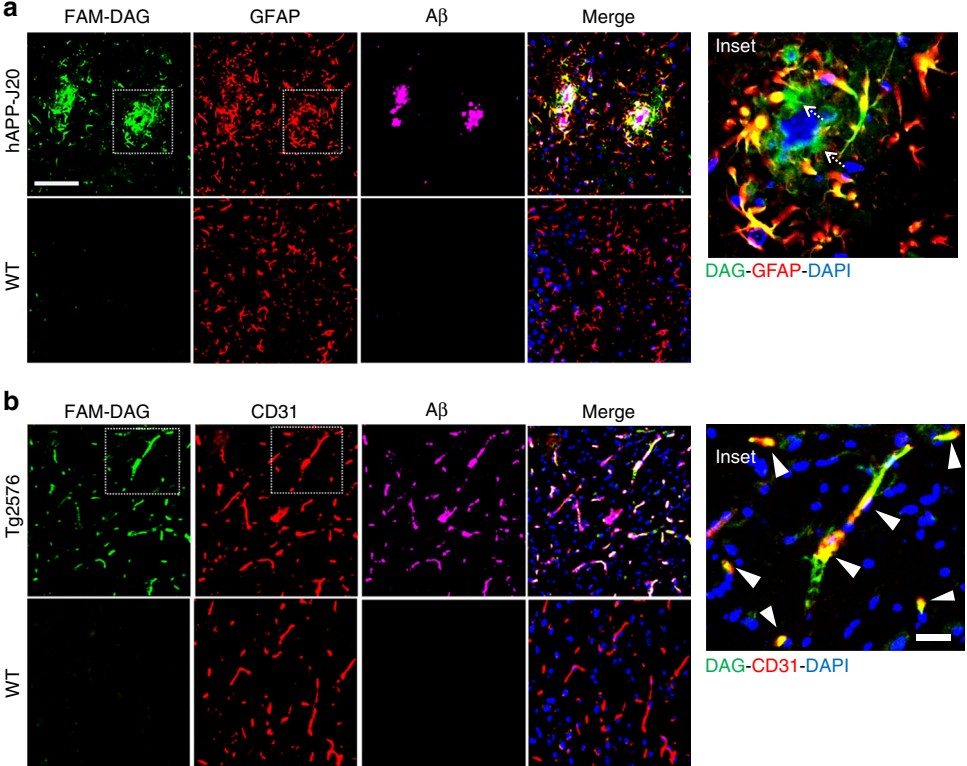

**Fig. 2** Homing of DAG peptide to the brains of AD model mice. **a** DAG targets activated astrocytes in AD mouse brain. **a** FAM-DAG was intravenously injected to 9-month-old hAPP-J20 and WT mice and allowed to circulate for 30 min. The mice were perfused, and the brains were fixed, sectioned and stained for FAM in DAG (*green*), GFAP (*red*) and Aβ (*magenta*). The region shown is the hippocampus. The *inset* shows co-localization between DAG and GFAP, and also DAG$^+$ and GFAP$^-$ regions (*arrows*). **b** DAG targets vessels positive for Aβ in Tg mice **b**. DAG was injected intravenously to 20-month-old Tg2576 WT mice, and processed as in panel **a**. The brain sections were stained for FAM (*green*), CD31 (*red*) and Aβ (*magenta*). The region shown is the cortex. The *inset* shows co-localization of DAG and CD31 (*yellow, white arrowheads*). *Scale bars*, 50 μm (**a** and **b**), 20 μm (*insets*)

peptides with the general composition of CX7C (C = cysteine; X = any amino acid) on the phage surface. The library was injected intravenously into the hAPP-J20 transgenic mouse model of AD[16] and their age-matched wild-type (WT) littermates in four different age groups (Fig. 1a). Four-fold higher phage titers were recovered from the hippocampi of 9-month-old hAPP-J20 than WT mice (Fig. 1b), whereas there was no significant difference in the younger age groups. These results suggest possible permeability of the BBB at 9 months of age, a time when the hAPP-J20 mice have fully developed disease.

High throughput sequencing analysis of the peptide-encoding region of the phage genome revealed peptide sequences that were highly enriched in the phage pools from the 9-month-old hAPP-J20 mice (Supplementary Fig. 1). Consensus motif analysis across all of the age groups further showed that one sequence motif, (A/G)BB(N/Q) (where B is a basic amino acid), was enriched in all hAPP-J20 age groups, but absent in the WT controls (Fig. 1c). One of the peptides in the 9-month phage pool CDAGRKQKC ("DAG") agreed with the consensus sequence and contained the most commonly occurring amino acids in the surrounding positions (Fig. 1c arrow). We focused subsequent analyses on the DAG peptide because it contained the motif that was present in peptides recovered from the hAPP-J20 mouse hippocampus at all stages of AD, from onset at 3 months through the late stage at 9 months of age (Fig. 1c).

Fluoresceinamine (FAM)-labeled synthetic DAG peptide homed from an intravenous injection to the brains of 9-month-old hAPP-J20 mice, but not WT mice. The homing was more prominent in the hAPP-J20 hippocampus than the cortex (Fig. 1d and Supplementary Fig. 2). There was little if any accumulation of

the peptide in other organs of the hAPP-J20 mouse except for kidney, probably due to the renal excretion of the peptide (Supplementary Fig. 3). A control peptide for DAG, with the same overall structure and charge (+2) (sequence: CRKQGEAKC) showed essentially no homing to the hippocampus of hAPP-J20 mice of the same age (Fig. 1d, bottom panel), thus, confirming DAG specificity for the hAPP-J20 brain. The same specificity of DAG for AD was observed in the Tg2576 model (Fig. 1e) as the peptide was detected both in the hippocampus and in the cortex (the region shown in the figure).

Further examination suggested that DAG labeled stellar-shaped glial cells adjoining blood vessels, and also partially co-localized with CD31-positive endothelial cell staining. The Tg2576 model showed more pronounced endothelial localization of DAG than hAPP-J20 (Figs. 1d, e, inset).

**DAG targets activated astrocytes in AD mice.** To identify the cellular target of DAG in adult transgenic AD mouse brain, we stained brain sections of DAG-injected hAPP-J20 mice with cell-specific markers for neurons (NeuN), microglia (Iba1) and reactive astrocytes (GFAP: glial fibrillary acidic protein). DAG did not show co-localization with neurons or microglia (Supplementary Fig. 4) but accumulated within a subpopulation of GFAP-positive hypertrophic astrocytes surrounding the Aβ plaques suggesting its specificity for activated (reactive) astrocytes (Fig. 2a). In AD, astrocytes are known to become reactive, particularly in the vicinity of amyloid plaques[17]. DAG accumulation correlates with astrocyte distribution, rather than that of Aβ (note the lack of DAG signal at the center of the Aβ plaque in

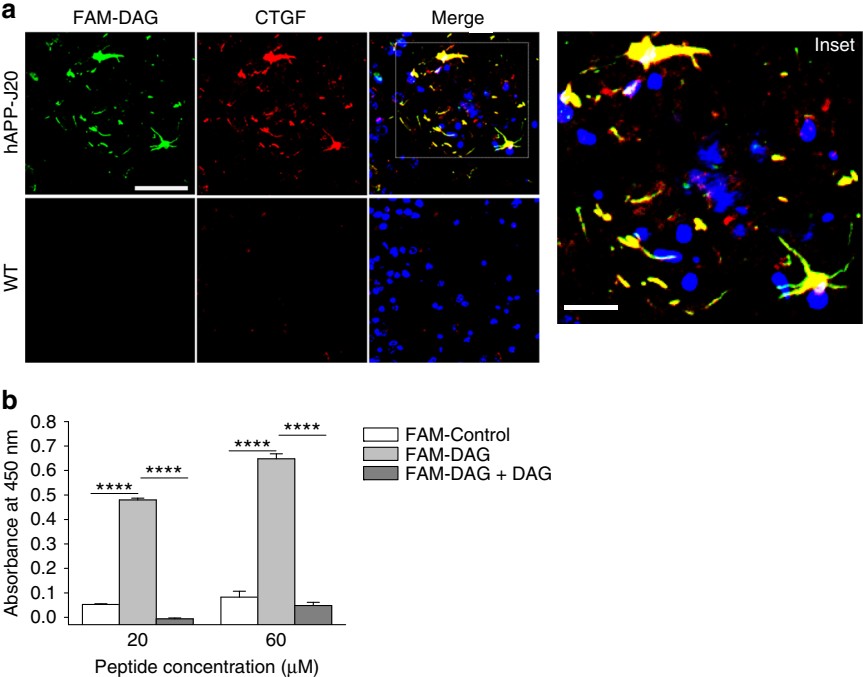

**Fig. 3** Identification of a receptor for DAG. **a** DAG colocalizes with connective tissue growth factor, CTGF. FAM-DAG was intravenously injected in 9-month-old hAPP-J20 and WT mice, allowed to circulate for 30 minutes, after which mice were perfused, and the brains were fixed, sectioned and stained for FAM (*green*) and CTGF (*red*). The region shown is the hippocampus. *Scale bars*, 50 μm (20 μm for *inset*). Representative images are shown. **b** DAG binds to recombinant CTGF. FAM-DAG was incubated on ELISA plate coated with recombinant human CTGF at two different peptide concentrations of the FAM peptide alone (*light gray bars*), or in the presence of non-labeled DAG (0.6 mM) (*dark gray bars*). The control peptide did not show significant binding to CTGF under the same conditions (*open bars*). Non-labeled DAG inhibited the binding of FAM-DAG, demonstrating specificity of the CTGF binding

panel a). In addition to astrocyte cell bodies, extracellular DAG accumulated in proximity to reactive astrocytes (Fig. 2a inset, white arrows). A similar pattern of both cellular and extracellular accumulation of DAG was observed in the brains of Tg2576 mice injected intravenously with FAM-DAG (Supplementary Fig. 5).

The affinity of DAG for hAPP-J20 mouse astrocytes was confirmed by ex vivo peptide binding to brain sections in an overlay assay. DAG bound to a much greater degree in the hippocampal region of hAPP-J20 mice than to age-matched wild-type mice (Supplementary Fig. 6). As observed for in vivo homing, DAG binding associated with GFAP positive astrocytes surrounding Aβ plaques.

**DAG targets endothelial cells in the cerebrocortex.** We further evaluated the strong vascular accumulation of DAG seen in the Tg2576 model. Immunostaining on brain sections of DAG-injected Tg2576 mice showed significant colocalization of DAG with Aβ and the endothelial cell marker CD31 (Fig. 2b inset, white arrowheads). Previous studies have shown that a subset of Aβ peptides localize to small blood vessels in AD brain, contributing to cerebral amyloid angiopathy (CAA)[18]. This phenomenon is reported to be particularly pronounced in aged Tg2576 mice[19]. Our data agree with those observations and show that DAG targets Aβ–positive vessels in these mice.

**DAG receptor identified as connective tissue growth factor.** To isolate the receptor for DAG, we identified an astrocytoma cell line (U251) that exhibited binding of DAG peptide. DAG-conjugated silver nanoparticles (DAG-AgNP) bound robustly to these cells, much more so than non-targeted nanoparticles (Figure S7). The binding of DAG-AgNP to the

U251 cells was specific, as it was inhibited by co-incubation with an excess of free, non-labeled DAG (Supplementary Fig. 7).

Next, we used U251 cell lysates for affinity chromatography separation on a DAG-affinity matrix (schematic in Supplementary Fig. 8). Mass spectrometry on the fraction eluted with free DAG peptide revealed a number of hits (listed in Supplementary Table 1) one of which was connective tissue growth factor (CTGF, also referred to as CCN2). CTGF is a member of the CCN family of matricellular proteins that is induced in inflammation and tissue repair[20]. We elected to focus initially on CTGF among the hits from the affinity chromatography because of the following considerations: (i) many of the candidate receptors were cytoplasmic proteins or had no reported or plausible connection to AD. (ii) CTGF is a heparin-binding matricellular protein, which would make it available for peptide binding in brain extracellular matrix and cell surfaces. (iii) Elevated expression of CTGF had been noted in small blood vessels of AD patient brains[21]. In addition, elevated CTGF expression takes place in various contexts of brain damage[22–24].

Next, we analyzed CTGF expression in mouse brains and noted significantly higher CTGF expression in hippocampus and cerebrocortex of hAPP-J20 and Tg2576 mice than WT mice (Supplementary Fig. 9). The elevated CTGF expression was associated with GFAP-positive astrocytes in the transgenic mice (Supplementary Fig. 10). In both AD transgenic mouse models, DAG strongly homed to CTGF-positive areas localizing in stellate shaped structures characteristic of astrocytes (Fig. 3a and Supplementary Fig. 11 respectively). The extracellular homing observed in the hippocampus in close proximity to reactive astrocytes we observed (Supplementary Fig. 5) is also in agreement with the notion that a matricellular protein such as CTGF is the DAG receptor.

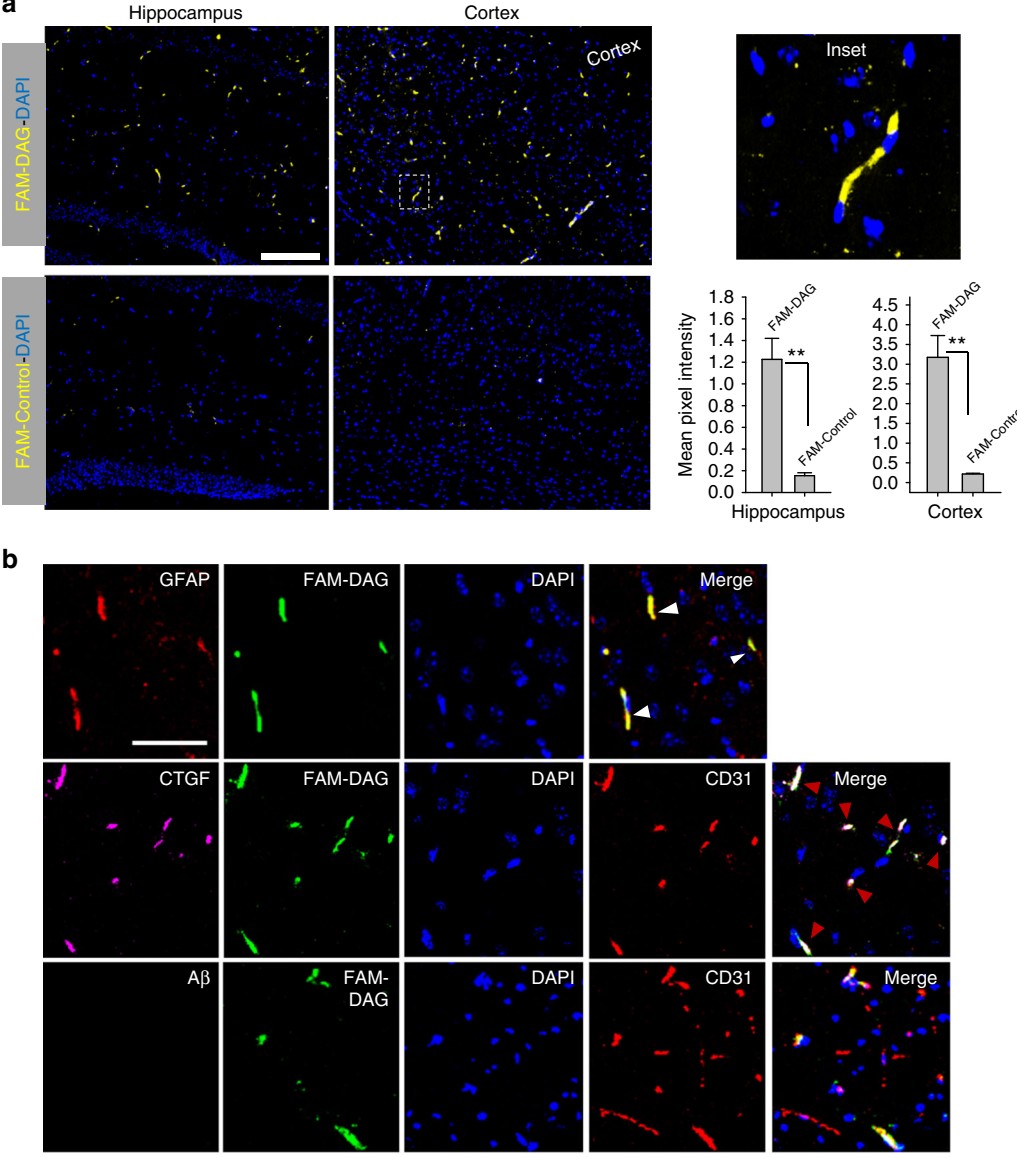

**Fig. 4** DAG homing to early stage hAPP-J20 mouse brain. **a** DAG localizes in the cerebrocortex and hippocampus of young (3–4 month old) hAPP-J20 mice after intravenous administration. FAM-DAG or control peptide were intravenously injected and allowed to circulate for 30 min. The mice were perfused, and brains were fixed, sectioned and stained for FAM (*green*). The signal from FAM was quantified using ImageJ and plotted in bar graph. The region shown is the cortex. **b** DAG co-localizes with CTGF and blood vessels in the hAPP-J20 mice. Sections from panel **a** were stained for FAM (*green*), CTGF (*magenta*), GFAP (*red* in first row), and CD31 (*red* in second and third row). The region shown is the cortex. *Arrowheads* point at DAG (FAM)-CTGF-CD31 co-localization. The blood vessels lack detectable Aβ accumulation (*bottom row*). Scale bar, 50 μm. **P < 0.01

To further validate CTGF as a DAG receptor, we tested in vitro binding of DAG to recombinant human CTGF. DAG bound to CTGF-coated plates in a dose-dependent manner (Fig. 3b). A control peptide showed only minimal binding, with no increase in binding with increasing peptide concentration. Furthermore, DAG binding to CTGF was inhibited in the presence of an excess of unlabeled DAG, confirming the specificity. These data indicate that CTGF is the DAG receptor in AD brain.

**DAG detects vascular changes in early AD in mice**. As the DAG peptide was originally identified from phage screening that spanned different ages of the J20 mice, we next tested DAG homing in hAPP-J20 mice with early-stage AD (3–4 months of age). We observed robust DAG homing in young hAPP-J20 animals, mostly in the cerebrocortex and the hippocampus

(Fig. 4a). At this age, DAG homing was predominantly found in blood vessels, co-localizing with CD31. Additionally, the peptide signal co-localized with GFAP-positive cells suggesting that it also targeted astrocytes in the neurovascular unit (Fig. 4b). The endothelial vessels were also positive for CTGF. Importantly, in agreement with prior reports, we did not observe any Aβ deposition at this early stage in this AD mouse model, which indicates that the homing of DAG to AD brain is independent of the presence of detectable amyloid deposits. Furthermore, DAG homing at this stage seems to be independent of BBB leakage, as no significant difference in the BBB permeability was observed between hAPP-J20 and WT mice (Supplementary Fig. 12).

**DAG specifically binds to human-derived AD samples**. Mouse AD models only go so far in mimicking the human disease, so it

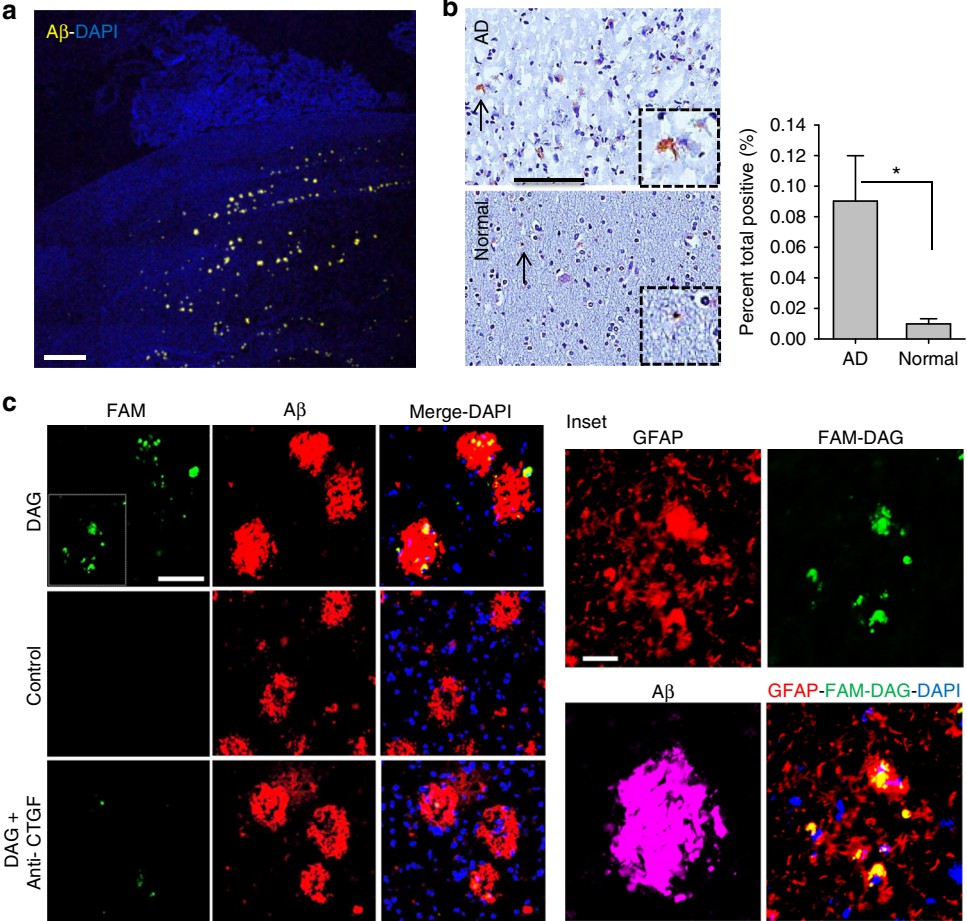

**Fig. 5** Aβ and CTGF expression in human brain, and DAG binding to human AD brain sections. **a** Sections from an AD patient brain were stained for Aβ (*yellow*) and counterstained with DAPI. *Scale bar*, 1 mm **b** Representative immunohistochemistry images of CTGF in brain sections of an AD patient and a control individual. The signal was quantified and plotted (*bar diagram*). *Scale bar*, 100 μm. *$P < 0.05$. **c** Human AD brain sections were incubated with either FAM-DAG (*top row*), control peptide (*middle row*) or anti-CTGF followed by FAM-DAG (*bottom row*), and stained for Aβ (*red*). Sections were counterstained with DAPI. *Green* signal is from native FAM fluorescence from peptide. *Scale bar*, 50 μm. *Inset* shows higher magnification of boxed area from panel c co-stained with GFAP antibody (*red*). *Scale bar*, 20 μm

was important to examine DAG for its ability to recognize human AD samples. We tested DAG binding to cultured BBB-type microvascular endothelial cells (BMECs) which had been differentiated from human induced pluripotent stem cells (hiPSCs) originally derived from AD patient fibroblasts (Supplementary Fig. 13a). DAG conjugated silver nanoparticles (DAG-AgNP) showed significant binding to the BMECS (Supplementary Fig. 13d). This binding was specific as it was inhibited by the presence of free DAG peptide (Supplementary Fig. 13e). We also tested ex vivo binding of DAG to sections from an AD human patient brain. DAG peptide bound strongly to the hippocampal region, which also displayed abundant Aβ oligomeric plaques (Fig. 5), whereas control peptide showed minimal binding. We also observed strong CTGF immunor-eactivity in the hippocampus in these sections (Fig. 5b). More-over, DAG binding to these sections was partially inhibited by pre-incubation with anti-CTGF antibody (Fig. 5c), confirming the specificity of DAG binding to CTGF. DAG binding to the human AD tissue coincided with regions with high GFAP expression (Fig. 5c inset).

**DAG-mediated payload delivery in AD**. To demonstrate the potential utility of DAG for site-specific delivery of payload into AD brain, we synthesized DAG conjugated iron oxide nanoparticles and tested the homing of intravenously adminis-tered nanoparticles to the brain of 9 month-old hAPP-J20 mice. We observed significant amount of nanoparticle delivery in the hippocampus and cortex of hAPP-J20 mice (Supplementary Fig. 14). No nanoparticle signal was observed in age-matched WT mice. In addition, our earlier data on delivery of FAM label on the DAG peptide and phage particles displaying DAG peptide further demonstrate the ability of DAG to carry a payload to AD brains.

**DAG homing in other models of neuroinflammation**. Since DAG targets activated astrocytes in AD, we tested in vivo homing of systemically injected DAG in other models of neuroin-flammation. Strong tumor homing of DAG was seen in P13 model of angiogenic glioblastoma, where the peptide co-localized with a subset of the GFAP-positive cells (Fig. 6a). Similarly, in a model of acute penetrating brain injury, DAG injected 6 h after injury robustly homed to the perilesional area containing activated hypertrophic astrocytes (Fig. 6b). Lastly, DAG also accumulated in GFAP-positive astrocytes in mTHY-1/α-synu-clein overexpressing mouse model of Parkinson's disease (Figs. 6c–e). Additionally, immunohistochemical analysis sho-wed significantly higher CTGF expression in brain sections from a patient with Parkinson's disease than normal brain (Supplementary Fig. 15). Collectively, these data suggest that

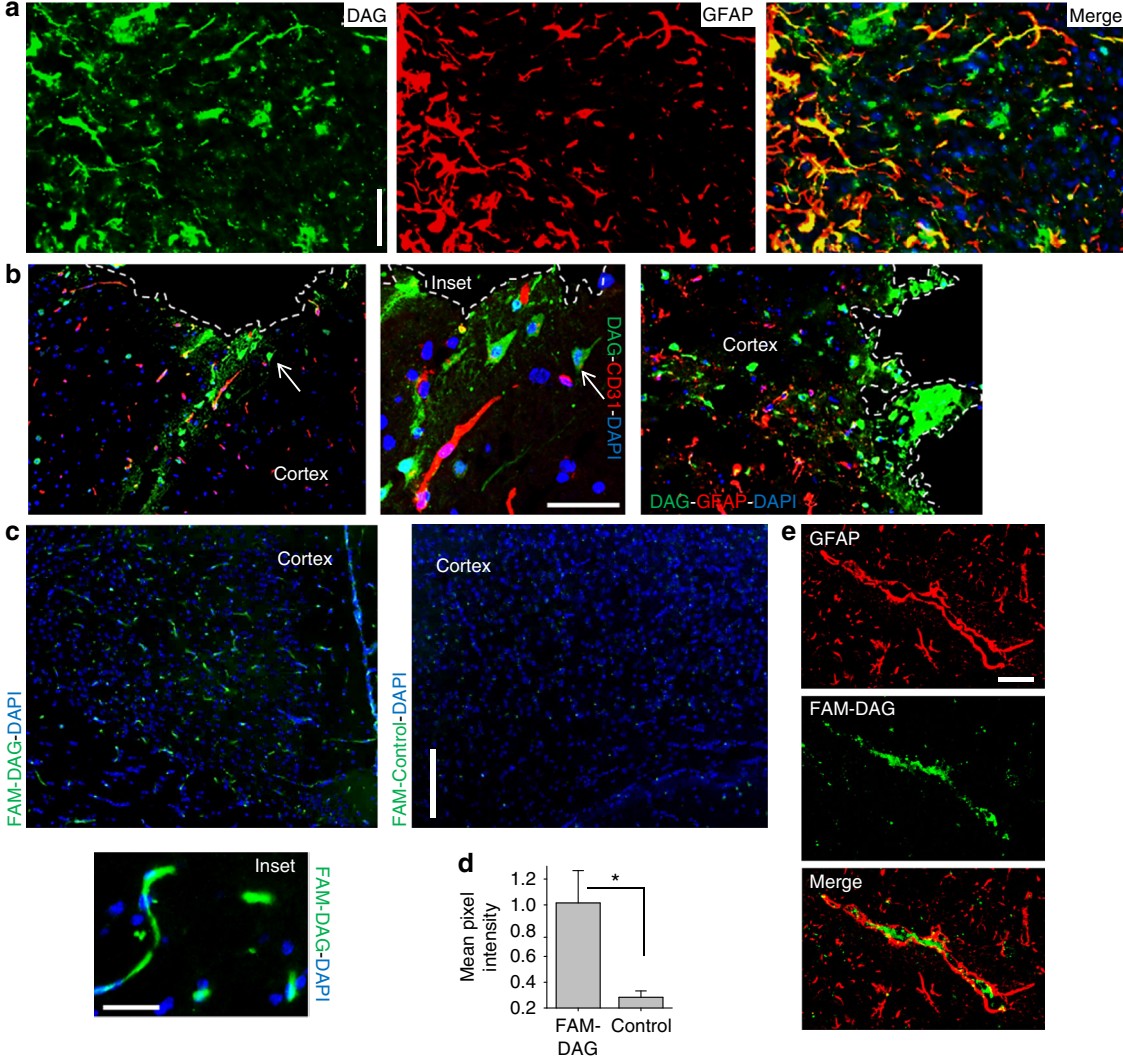

**Fig. 6** DAG homes to the brain in mouse models of neuroinflammation. **a** DAG targets brain in mouse model of glioblastoma. FAM-DAG was intravenously injected to mice bearing an angiogenic glioblastoma derived from a patient (P13), and allowed to circulate for 30 min, after which the mice were perfused, the tumors were fixed, sectioned and stained for FAM (*green*), and GFAP (*red*). Scale bar, 40 μm. **b** DAG homes to acute brain injury in mice. FAM-DAG was injected intravenously in mice with penetrating brain injury, processed as in panel **a**. Representative immunofluorescence images of brain sections from the injury area stained for FAM (*green*), and CD31 (*red; left and middle panels*) or GFAP (*red; right panel*) show stellate-shaped cells positive for DAG (see *arrow*). Scale bar, 40 μm. **c**–**e** DAG homes to the brain in a Parkinson's disease mouse model. FAM-labeled DAG or control peptide was intravenously injected to Parkinson's disease mice (mTHY-1-α-synuclein model, 1-year old) and processed as in panel **a**. **c** Brain sections were stained for FAM (*green*) and counterstained with DAPI Scale bar, 100 μm. *Inset* shows a typical worm-like structure targeted by DAG in the cortex. Scale bar, 20 μm. **d** Fluorescence signal from FAM in panel **c** was quantified using ImageJ. **e** Brain sections from DAG-injected mice were stained for FAM (*green*) and GFAP (*red*). DAG signal associates with activated astrocytes. Scale bar, 40 μm. *$P < 0.05$

DAG targets activated astrocytes in both acute and chronic models of neuroinflammation.

## Discussion

We describe the application of in vivo phage display in mouse models of AD for identification of peptides that target components of the neurovascular unit. One of the peptides, DAG, selectively recognizes a subset of astrocytes that are activated in AD starting at an early stage of the disease. We also show that the peptide recognizes activated astrocytes in other models of neuroinflammation. Importantly, the target of this peptide is expressed both in mouse and human AD brain.

To our knowledge, this is the first application of in vivo phage display to probe for specific, systemically accessible molecular

signatures present in the AD brain. The higher number of phage recovered from brains of aged hAPP-J20 mice compared to age-matched normal mice and younger hAPP-J20 mice is consistent with the notion of a leaky BBB in late stages of the disease. Vascular impairment represents an important factor in the pathology of AD, particularly with dysfunction of components of the neurovascular unit, and disruption of the BBB has been widely reported in animal models of AD, as well as patients with late-stage AD[25, 26]. Increased permeability of the BBB apparently allows the phage to enter the subarachnoid space, which is normally not accessible, resulting in some phage binding to specific extravascular targets. The DAG peptide identified in our screen shows binding and accumulation in brain endothelia as well as in the brain parenchyma. Binding of the DAG peptide to components of the neurovascular unit (endothelial cells and astrocytes)

in AD brain is of particular interest because the vasculature can be readily accessed through the systemic circulation. However, while DAG accumulated in the brain endothelial cells and astrocytes in both models we tested, the extent of accumulation in these cell types was different in the two models. It is difficult to suggest possible reasons for the difference because the two models differ from one another in a number of ways, including the APP mutation, the promoter driving the APP production, and the age of onset of the disease.

The homing pattern of DAG also differed depending on the stage of AD. In the hAPP-J20 model, the homing was vascular at the early stage of AD and extravascular later on. The reason for this difference could be limited accessibility through an intact BBB at the early stage, or a change in expression pattern of the DAG receptor at different stages of AD, or both. Vascular alterations in AD have been reported, even at early stages of the disease[3, 4, 26]. Moreover, early in the disease, neuroinflammation and synaptic/neuronal loss precede Aβ plaque deposition and tau tangle formation, hallmarks of late stage AD[27]. Our data showing DAG homing to early stage AD in the hAPP-J20 mouse model indicate that the peptide recognizes an early change in the neurovascular niche.

It was important to rule out the possibility that BBB leakage was the reason for the in vivo brain accumulation of DAG in the AD mice. We found that the binding of the peptide to brain sections was independent of access through the BBB. DAG binding to hAPP-J20 brain sections was far stronger than DAG binding to WT brain or a control peptide binding to hAPP-J20 brain. These results suggest that upregulation of the DAG target molecule is a factor in the preferential homing of systemically injected DAG to AD brain. While BBB permeability is likely needed for DAG to reach extravascular targets, the blood vessels should be available for DAG binding regardless of the status of the BBB. Our results on the DAG target molecule discussed below confirm these conclusions.

We provide evidence that CTGF is the target molecule for DAG. CTGF is a matricellular protein that acts as a regulator of several cellular functions, including cell adhesion, migration, mitogenesis, differentiation, and survival[28]. CTGF was one of several candidate proteins that we identified from DAG affinity chromatography, and we focused on it as a candidate receptor for DAG because high CTGF expression has been reported in activated astrocytes in the brains of human AD patients[21]. We also found high CTGF expression in AD brains, both human and mouse. In addition, we found that the endothelial cells and adjoining astrocytes in hAPP-J20 mouse brain were positive for CTGF immunostaining, in agreement with the observation of accumulation of systemically injected DAG in these cells. Further evidence for CTGF as the DAG receptor includes direct binding of FAM-labeled DAG to recombinant CTGF and inhibition of the binding with unlabeled DAG. Moreover, the binding of CTGF to extracellular matrices creates an insoluble fraction of the growth factor that can immobilize the peptide binding to it. Based on these data, we conclude that CTGF is the receptor for DAG. Future studies are needed to address the possible effects of DAG binding on CTGF function.

DAG seemed to colocalize with Aβ in late stage AD, raising the question as to whether DAG might bind to Aβ. However, CTGF is secreted by reactive astrocytes[21], the main target of DAG homing, and reactive astrocytes surround Aβ deposits, creating an impression of DAG colocalization with Aβ. Closer examination showed that DAG is associated with the astrocytes and not Aβ. Moreover, DAG homing in models with no Aβ deposition such as early stage hAPP-J20 mice, and Parkinson's and glioblastoma mice indicates that Aβ deposition is not necessary for DAG

homing. Rather, DAG homing seems to be only dependent on astrocyte activation and CTGF expression.

Since we initially identified the DAG peptide using mouse models of AD, it was important to determine if the peptide also recognizes the human AD tissue. Our results show that in fact it does. First, DAG binds to human CTGF, as shown by the binding of DAG to the U251 human astrocytoma cells we used as the source of stating material for the identification of CTGF as the DAG receptor. Second, the recombinant CTGF that we used to show the binding of DAG to CTGF was of human origin. Third, DAG specifically bound to human AD iPS cells differentiated into BBB-type endothelial cells. Finally, DAG bound to human AD brain sections in an overlay assay. Taken together with the demonstration of elevated CTGF expression in human AD brain, these results show that DAG is relevant to human AD.

Studies have reported CTGF up-regulation in patients with conditions other than AD that have a neuroinflammatory component, such as Parkinson's disease[29], brain injury[24], glioblastoma[30], and cerebral infarction[31]. Given its potential to modulate the cellular phenotype and remodel tissue in the CNS following injury and in neurodegenerative disease, CTGF may represent an attractive therapeutic target. The fact that DAG homes to brain in animal models of Parkinson's disease, brain injury, and glioblastoma agrees with the published CTGF expression pattern and suggests that DAG has the potential for broad applicability in brain diseases. Our data on DAG mediated delivery of nanoparticle payload to AD brains supports the concept of utilizing DAG for imaging applications, in which the differential diagnosis could be based on the pattern of accumulation of the DAG-guided contrast agent. A particularly attractive potential application of DAG would be as a biomarker for early detection of AD, as suggested by DAG homing to AD brain vessels at early stages of disease in the hAPP-J20 mouse model.

In conclusion, the DAG peptide provides a tool for targeting of the neurovascular unit to improve imaging and management of neuroinflammatory conditions. It also draws attention to the potential role of CTGF in AD and other neurological diseases, which heretofore has been essentially unexplored.

## Methods

**Animal models**. All animal experiments were conducted under an approved protocol of the Institutional Animal Care and Use Committee of Sanford Burnham Prebys Medical Discovery Institute. The following transgenic mouse models were used for peptide homing studies - hAPP-J20[16], Tg2576[32], and mTHY-1-α-synuclein model[33]. All three models are derived from *C57BL/6* mice and were used at ages described in the specific experiment. The P13 glioblastoma model is a patient-derived orthotopic xenograft model[34] received from Dr. Rolf Bjerkvig´s lab (NORLUX Neuro-Oncology, Department of Biomedicine, University of Bergen, Norway). Spheroids were cultured as described previously[35]. After two weeks in culture, spheroids (volume 2.5 µl) were stereotactically implanted into the brains of nude mice. Mice were used approximately 6 weeks after tumors were induced, when mice started presenting neurological symptoms. The acute brain injury model was setup as previously described[15]. Briefly, 8 to 10 week-old male BL6 mice were anaesthetized with 4% isoflurane in 70% $N_2O$ and 30% $O_2$, and a 5-mm craniotomy was performed using a portable drill and a trephine over the right parietotemporal cortex using a stereotactic frame. The bone flap was removed and nine needle punctures using a 21 G needle were made 3 mm deep according to a $3 \times 3$ grid, spaced 1 mm in width and 1 mm in height. The scalp was then closed with sutures, anesthesia discontinued and mice were administered buprenorphine i. p. for pain control.

**In vivo phage display**. The in vivo phage screening in hAPP-J20 mice was done as described[12]. A CX7 C naïve phage library (1e10 pfu of, in 100 µL of PBS) was intravenously injected and allowed to circulate for 30 minutes due to its short halflife, after which mice were anesthetized with 2.5% avertin and perfused with PBS intracardially. The brain was removed and the hippocampus was extracted and homogenized in LB-NP 40 (1%) and phage was processed as described[12]. The phages in the lysate were rescued by amplification in *E.coli* and peptide-encoding portion of the phage genome was sequenced using Ion Torrent high throughput sequencing.

**Homing studies and tissue sections**. For peptide homing, mice were intravenously injected with 50 nmoles of peptide dissolved in PBS, and allowed to circulate for 30 min. Mice were then perfused intracardially with saline and all major organs were isolated and fixed in 4% paraformaldehyde (PFA) at pH 7.4 overnight. The organs were then washed with PBS and placed in graded sucrose solutions overnight before optimal cutting temperature compound (OCT) embedding. Ten-micro-meter-thick sections were cut and analyzed by immunostaining.

**Peptide synthesis and coupling**. The peptides were synthesized on a microwave-assisted automated peptide synthesizer (Liberty; CEM, Matthews, NC) following Fmoc/t-Bu (Fmoc:Fluorenyl methoxy carbonyl, t-Bu: tertiary-butyl) strategy on rink amide resin with HBTU (N,N,N′,N′-Tetramethyl-O-(1H-benzotriazol-1-yl) uranium hexafluorophosphate (OR) O-(Benzotriazol-1-yl)-N,N,N′,N′-tetra-methyluronium hexafluorophosphate) activator, collidine activator base and 5% piperazine for deprotection. Fluorescein and biotin tags were incorporated during synthesis at the N-terminus of the sequence. Cleavage using a 95% TFA Trifluoro acetic acid followed by purification gave peptides with > 90% purity. Peptides were lyophilized and stored at -20 °C.

**ELISA**. To demonstrate the specific binding of DAG peptide to its receptor, ELISA plate wells were coated with full-length recombinant human CTGF/CCN2 (R&D systems; cat#9190-CC) at a concentration of 5 µ/ml for 2–3 h at room temperature. The wells were washed three times with PBS/0.05% Tween-20 ("PBST"), and blocked with 1% (w/v) Casein in PBST (Thermo Scientific; cat# 37582) for 2 h at room temperature. 5(6)-carboxyfluorescein label (FAM)- DAG or the control peptide diluted in 0.05% Casein/PBST were added to the respective wells and incubated overnight at 4 °C with shaking. For competition experiment, excess of unlabeled DAG peptide (0.6 mM) was added in the solution. After washing wells one time with PBS (containing 300 mM NaCl) and twice with PBST, 100 µl of anti-FITC HRP conjugated polyclonal antibody (GeneTex #GTX26656) diluted (1:10,000) in 0.05% Casein/PBST was added to the wells and incubated with shaking for 30 min at RT. The wells were finally washed there times with PBST and plate was developed with TMB substrate (3, 3′, 5, 5′-Tetramethylbenzidine) (Sigma) for 20 min. The reaction was stopped with the addition of equal volume of 1 M sulphuric acid and absorbance measured at 450 nm (FlexStation 3 Reader, Molecular Devices, Sunnyvale, CA, USA).

**Tissue section overlays of FAM labeled peptides**. PFA fixed sections were incubated in PBS for 10 minutes, followed by PBS-Triton-x-100 (0.2%) for 10 minutes. Sections were rinsed with PBST, mounted in a slide holder, and subsequently washed with PBST (3 washes of 4 minutes each). Slides were blocked with blocking buffer for 60 minutes at room temperature. Then, peptides were incubated at a concentration of 10 µg/mL dissolved in concentrated blocking buffer, for three hours at room temperature. Sections were then washed with three 4-minute washes of PBST, and subsequently fixed with PFA for 10 minutes, followed by three 3-minute washes with PBST. Sections were then incubated with primary antibody at 1/200 in diluted blocking buffer overnight at 4 °C. Later, sections were washed with three 4-minute washes with PBST. Slides were then incubated with secondary antibody at 1/200 dilution in diluted blocking buffer for 30 minutes at room temperature, followed by three 4-minute washes of PBST. Sections were then counterstained with DAPI for 5 minutes, washed with PBST and mounted.

**Immunofluorescence**. Frozen sections were permeabilized using PBS- 0.2%Triton X-100, blocking was carried out using blocking buffer (5% BSA, 1% goat serum, 1% donkey serum in PBST). Primary antibodies were incubated in diluted (1%) blocking buffer overnight at dilutions 1/100 or 1/200 at 4 °C, washed with PBST and incubated with secondary antibodies diluted 1/200 or 1/500 in 1% diluted buffer for 1 h at room temperature, subsequently washed with PBST, counterstained with DAPI 1 µg/ml in PBS for 5 minutes, washed with PBS, mounted using aqueous mounting media (Vector Biolabs), and imaged using a confocal microscope (Zeiss LSM-710). Staining was done using the following antibodies and reagents: anti-fluorescein (Invitrogen A889; 1:100), CD31 (BD Biosciences clone MEC 13.3; 1:50), aβ (Sigma clone 6E10; 1:200), GFAP (Invitrogen clone 2.2B10; 1:400), CTGF (Santa Cruz, clone C-19; 1:100), Iba 1 (Wako Chemicals, 019-19741; 1:200), NeuN (Abcam, ab177487, 1:200).

**Affinity chromatography and proteomics**. For identifying DAG binding proteins, the human glioblastoma astrocytoma cell line U251 was lysed in PBS containing 200 mM n-octyl-beta-D-glucopyranoside and protease inhibitor cocktail (Roche) as described previously[15, 36] with slight modifications. The clarified lysates were loaded on to Sera-Mag magnetic particles (GE Healthcare Lifesciences, USA) coated with biotin-DAG, and incubated overnight with rotation at 4 °C. The magnetic beads were washed with wash buffer followed by additional washing with 0.5 mM control peptide (CRKQGEAKC) to remove non-specifically bound proteins. The bound proteins were eluted with 1 mM free DAG peptide. The eluted fractions were pooled, their protein concentration determined by using bicinchoninic acid (BCA) protein assay (Pierce) and the samples were digested using the Filter-aided Sample Preparation (FASP) method[37]. Finally, the digested samples

were dried, desalted and subjected to LC-MS/MS analysis at the Sanford Burnham Prebys Medical Discovery Institute's Proteomics Core facility. All mass spectra were analyzed with MaxQuant software version 1.5.0.25. The MS/MS spectra were searched against the Uniprot protein sequence database (version July 2014). The proteomics data and detailed method has been deposited to MassIVE repository (accession MSV000081390).

**Generation of brain microvascular endothelial cells from hiPSCs**. Human induced pluripotent stem cells (hiPSCs) were generated following approval from the Stem Cell Research Oversight committee of Sanford Burnham Prebys Medical Discovery Institute. The details of hiPSCs derived from Alzheimer disease patients with APP duplication, and non-demented control individuals, are described previously[38]. These hiPSCs were extensively characterized and have been established as an excellent human model for AD. An additional AD hiPSC line generated from dermal fibroblasts from a 56-year-old individual harboring a Presenilin1 (PSEN1) mutation (Coriell Institute, Cat # AG06840) was used to further validate the results from the APP lines. We routinely maintain hiPSCs on mouse embryonic fibroblast as described before[39, 40]. Brain microvascular endothelial cells differentiation of hiPSCs was performed using a previously described protocol[41], with minor modifications. Briefly, feeder-free cultures of hiPSCs were allowed to spontaneously differentiate in the absence of bFGF for 5–7 days, and then transferred to Endothelial cell (EC) medium composed of human Endothelial Serum-Free Medium (Invitrogen) supplemented with 20 ng/ml bFGF and 1% platelet-poor plasma-derived bovine serum (PDS; Biomedical Technologies). After 1–2 days the cells were dissociated with dispase (2 mg/ml) and were plated on 12 well plates coated with a mixture of collagen IV (400 µg/ml) and fibronectin (100 µg/ml). Cells were then cultured in EC medium until they reached confluence, after which they were split and expanded to near 100% purity. BBB-type EC identity was confirmed by double immunoreactivity to the hallmark efflux transporter p-glycoprotein and other EC markers (CD31, GLUT-1, PECAM, Occludin, and Claudin-5). Furthermore, the capability of these cells to make functional tight junctions and polarized efflux activity was validated using a dual chamber efflux transport assay as described before[41].

**Nanoparticles synthesis and peptide conjugation**. Silver nanoparticles (AgNPs) with PEG coating and peptide functionality were prepared as reported previously with some modifications[42]. AgNPs of ~35 nm diameter were synthesized by citrate acid reduction of silver nitrate in solution[43]. First, AgNO$_3$ (450 mg) dissolved in 2.5 L water was stirred and heated to a boil and 50 mL water containing trisodium citrate dihydrate (500 mg, Sigma) was added. After 30 min the solution was cooled to room temperature. The resulting optical density at 405 nm was ~10. To install the coating, lipoic PEG amine (LPN, 51.9 mg, 3400 g/mol, Nanocs) was dissolved and reduced for 3 h in 4.1 mL of aqueous 84 mM tris-carboxylethyl phosphine pH 7.0 (Sigma). AgNPs (500 mL) were heated to 50 °C and LPN solution (0.79 mL) was added, followed by 0.25 mL of 0.5 M TCEP. After 30 min incubation the solution was cooled to room temperature (RT) forming LPN-AgNPs. Tween 20 (T20, 0.25 mL, 10% in water) and 20 mL 2 M NaCl were sequentially added to the LPN-AgNPs and incubated overnight at 4 °C. Using a stirred cell apparatus (Millipore) equipped with a 100 kDa membrane LPN-AgNPs were washed and concentrated 50-fold into 0.5X PBS with 0.005% T20 and 5 mM TCEP. LPN-AgNPs were further passivated by adding 0.03 mM N-acetyl-L-cysteine methyl ester (Sigma) for 2 h, followed by 0.10 mM tetracysteine peptide (acetyl-CCPGCC-amide, LifeTein) for 2 h. LPN-AgNPs were washed twice at 15k RCF and resuspended to 400 O.D. (optical density) in 0.05 M phosphate buffer with 0.005% T20 pH 7.3. This product could be stored at least 6 months at 4 °C. To attach peptide, a bifunctional linker was reacted with 1 mL of the LPN-AgNPs to introduce maleimide groups (10 mg, NHS-PEG-Mal, 2 kDa JenKem USA, 1 h at RT), then washed with 0.1 M HEPES buffer pH 7.2 0.005% T20 by centrifugation (4 C, 11 kxg 15 min, three times), and immediately reacted for 1 h with freshly dissolved cysteine peptide (final concentration ~0.1 mM FAM-cys-x-DAG-NH$_2$) or a control thiol-containing peptide. X indicates aminohexanoic acid linker. The product peptide-AgNPs were washed with PBS 0.005% T20 (PBST), then filtered (0.22 µm). The Ag plasmon peak was 300 O.D. at 405 nm. We estimated ~15 nM in AgNPs using an extinction coefficient of $2 \times 10^{10}$ M$^{-1}$ cm$^{-1}$ for spherical silver obtained from[44]. The synthesis and subsequent conjugation of iron oxide nanoparticles (termed nanoworms (NW) has been described[45]. For peptide coupling, aminated NWs were PEGylated with maleimide-5KPEG-NHS (JenKem Technology) and DAG peptide containing an extra cysteine was conjugated to the functionalized particles through a thioether bond between the cysteine thiol in the peptide and the maleimide on the NW.

**In vitro binding experiments**. Cell binding experiments on U251 cells and hiPSCs were done using peptide conjugated AgNP. U251 cells were cultured in a 96-well plate. The cells were blocked with 200 µl of 10% FCS in HBSS (Hanks' Balanced Salt Solution from Gibco) for 30 min at 37 °C. Following that, DAG-AgNPs (0.5 nM concentration diluted in HBSS) alone or in the presence of free non-labeled peptide (200 µM) were incubated on the cells for 1 h at 37 °C. After washing the unbound AgNP with HBSS three times, the plates were imaged with fluorescent microscopy by looking at intrinsic emission from the FAM tag on the

peptide. hiPSCs were cultured in 48 well plate and similar protocol as described above was followed, with the exception that AgNP concentration for incubation was kept at 0.15 nM. Nanoparticle binding was quantified from fluorescence micrographs using ImageJ software.

**Human tissue experiments**. Postmortem human brain tissue was obtained from the New York Brain Bank at Columbia University, New York. The patient was an 82-year-old male with a diagnosis of Alzheimer's disease neuropathologic changes ([A3, B3, C3]). The normal brain tissue was obtained from BioChain Institute Inc. (Newark, CA). The donor was a 54 year-old male without any neurologic diagnosis on detailed neuropathologic evaluation. The Parkinson's disease brain tissue was obtained from UCSD Pathology bank. Informed consent was obtained for all autopsies. Frozen brain tissue was sectioned for immunohistochemistry and ex vivo overlay binding with FAM-peptides.

**Statistical analysis**. All data represents mean value ± SEM. All the significance analysis was done using Statistica 8.0 software, using one-way ANOVA or two-tailed heteroscedastic Student's $t$ test. The details of the statistical tests carried out are indicated in respective figure legends.

**Data availability**. All data needed to evaluate the conclusions in the paper are present in the paper and/or the Supplementary Materials. Additional data related to the findings of this study are available from the corresponding author.

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

## Acknowledgements

We thank Prof. R. Bjerkvig at University of Bergen for sharing the P13 glioma cells, Prof. Lawrence Goldstein at UCSD for AD hiPSCs; Nima Dolatabadi, Daniel Joseph and James Parker at the Scintillon Institute for assistance in studies involving hiPSC differentiation; Tracy Fang Newmeyer for help with animal studies; Venkata R. Kotamraju for peptide synthesis; and the New York Brain Bank at Columbia University for sharing human AD brain tissues. Proteomics analysis and histology was performed by proteomics and histology core respectively at Sanford Burnham Prebys, supported by the NCI Grant, P30 CA30199. This work was supported in part by the Defense Advanced Research Projects Agency (DARPA) under Cooperative Agreement N66001-14-1-4010 (to E.R.). The findings and views expressed are those of the authors and do not reflect the official policy or position of the Department of Defense or the U.S. Government. T.T. was supported by grants from the European Research Council (No. 291910), the Wellcome Trust International Fellowship (WT095077MA), and by European Union through the European Regional Development Fund (Project No. 2014-2020.4.01.15-0012). R.A. and S.A.L. were supported by NIH grants P01 HD29587, R01 NS086890, DP1 DA041722, and P30 NS076411, and by a Distinguished Investigator Award of the Brain & Behavior Research Foundation.

## Author contributions

A.P.M., P.S., S.A.L., E.R. designed the experiments. A.P.M., P.S., S.H., T.M., K.T., performed the research; R.A. and G.B.B. contributed reagents or developed cell-based models; A.P.M., P.S., T.T., S.A.L., E.R. analyzed the data and edited the manuscript.

## Additional information

**Competing interests:** A.P.M., P.S., S.H. and E.R. are inventors on the patents. A.P.M., S.H. and E.R. have ownership interest in AivoCode and are founders and officers of the company. The remaining authors declare no competing financial interests.

