## [Peer Review File · Nature Communications]

Reviewers' comments:

Reviewer #1 (Remarks to the Author):

In this manuscript, using in vivo phage display approach, Mann AP et al identifies a "DAG" peptide that accumulates in endothelial cells and reactive astrocytes of Alzheimer's disease mouse brains. They further suggest that the DAG targets CTGF, a matricellular protein highly expressed in mouse and human AD. In addition, the authors show that DAG homes to cells and brain tissues of several other disease models, in which CTGF is expressed. Overall, the study is interesting with comprehensive analysis. However, there are a number of important issues that need to be addressed in the manuscript as outlined below.

1. How do the authors quantitate the affinity of FAM-DAG to vascular units (such as in endothelial cells or astrocytes)? In Fig 1d, the FAM-DAG scatteredly appears in cells positive for CD31 in hippocampus of hAPP J20 mice (1 out of 7 CD31+ cells co-localizes with FAM-DAG). In Fig 1e, Fig 2 and S3, however FAM-DAG accumulates in endothelial or astrocytes at different extent.

2. In Fig 1d and e, and Fig S3, why FAM-DAG preferentially locates in hippocampus of hAPP-J20 while it appears to wildly locate in Tg2576 mice. How do the authors explain this brain region preference of DAG peptide? In Fig S3, the FAM-DAG peptide clumps in Tg2576 mouse brains even at the place of which Ab plaques are absent, and reactive astrocytes only locate at the border zones of these clumps. Does it mean that FAM-DAG in fact has other targets or aggregates?

3. The authors mentioned that CTGF was one of several candidate proteins identified from DAG affinity chromatography. How about other candidates? Are they all located in vascular units? What is the specificity of DAG to vascular units?

4. In Fig 3, a negative control is needed before the authors conclude that DAG is specific to CTGF and that CTGF is a DAG receptor. Does DAG bind to other members of CCN family proteins or other proteins in astrocytes?

5. Authors need provide data confirming the cells they differentiated from hiPSCs are the BMECs. Also, data are needed to support their claims that DAG-AgNPs bind to the BMECs from hiPSCs. From the Fig S10, one cannot judge which cell types the DAG-AgNPs bind to, and the cells are not labeled by any markers.

6. In Fig 5B, the images of CTGF immunostaining are not consistent with the histogram; the imaging does not reflect about 7-fold increase in the CTGF expression as shown in the histogram.

7. Authors described that DAG binds to GFAP+, CD+ cells which are positive for CTGF (Fig. 4B) at the early-stage of AD (3-4 months of age), the time point which Abeta deposition is absent (page 8). However, in human AD (Fig 5C) and mouse AD (Figs. 2 and S3), the authors consistently show that DAG is co-localized with Abeta plaques. Does DAG itself have affinity with Abeta polypeptides? Or, the colocalization between DAG and Abeta plaques is due to the affinity of DAG with reactive astrocytes surrounding the Abeta plaques. This needs to be clarified, as different molecular action of DAG might be involved.

8. Though reactive astrocytes suggest neuroinflammation, the data shown by the authors mainly indicate that the DAG peptide binds to the components of the neurovascular unit, likely reflecting vascular impairment or alteration during AD. Maybe, it is a similar case to PD (the image of PD mouse brains in Fig 6 looks like that FAM-DAG labels vessels). Thus, the authors need more evidence to

validate DAG-neuroinflammation aspect of this work, before bringing DAG to a broad indication.

Minor comments:

9. The authors i.v. injected a CX7C naïve phage library for 30 min in animal models and then harvest tissues to search peptides. Why do they use 30 min, is there any specific reason or rationale for that? There is no explanation on it through the manuscript.

10. Page 5 line 9, the authors did not indicate the figure number regarding the results they described.

Reviewer #2 (Remarks to the Author):

In this manuscript the authors used in vivo phage display of a random peptide library in the hAPP-J20 transgenic mouse model of amyloidosis at four different ages and identified a cyclic peptide, CDAGRKQKC ("DAG") that accumulated in the hippocampus of the hAPP-J20 mice. The DAG peptide homed to endothelial cells and activated astrocytes in the hippocampus of both hAPP-J20 and Tg2576 mouse models of amyloidosis and its target was identified as connective tissue growth factor (CTGF), a protein highly expressed in mouse and human AD. DAG targeted various ages (3-9 months) and localized in hippocampus and cortex of hAPP-J20 mice even prior to amyloid deposition especially in blood vessels. DAG also showed high binding to hiPSCs derived from an AD patient. DAG also homed to cells in a glioblastoma model, in a model of penetrating brain injury, and a model of Parkinson's disease brain, all of which express CTGF. The authors conclude that DAG thus provides a tool for targeted delivery of therapeutics and imaging agents into neuroinflammatory lesions, with implications for pathogenesis of AD and additional neurodegenerative disorders with a neuroinflammatory component.

The manuscript is interesting and provides a potential new target for studying neuroinflammation. However, the conclusions regarding the potential use of DAG to deliver drugs or as a marker are only speculative and not supported by any data. Overall the manuscript describes the identification and distribution of DAG in various disease models but no evidence that it is actually useful. I have one major comment and two minor comments to improve the manuscript:

Major comment: It would greatly improve the manuscript if the authors could show that DAG can be used to deliver any attached molecule of their choosing (drug, radiographic label etc.) in any of the models of neuroinflammation used in the manuscript, in order to demonstrate the usefulness of the peptide.

Minor comment #1: the authors liberally use the words Parkinson disease, TBI, Glioblastoma while referring to far from perfect mouse models of these diseases giving the impression that the peptide homed on cells in the actual diseases (not in the models). This should be made clear especially in the abstract where the authors say: "DAG also homed to cells in human Glioblastoma, Traumatic brain injury and Parkinson's disease brain". The text should read "DAG also homed to cells in mouse models of glioblastoma.....".

Minor comment #2: It is not clear how the findings of the manuscript have implications for pathogenesis of Alzheimer's disease and other neurodegenerative diseases. This should be further elaborated upon in the text -or the sentence removed from the abstract.

Reviewer #3 (Remarks to the Author):

1. Comments for Author

The purpose of this study by Mann, Scodeller, et al. was to identify peptides via in vivo phage display that specifically recognize molecular changes associated with Alzheimer's disease (AD) pathogenesis and enable targeting of these sites via systemic administration. In order to accomplish this, the authors injected a T7 phage library intravenously into hAPP-J20 mice of different ages. The authors then examined the hippocampi of the injected animals for highly enriched peptide sequences and identified a peptide (named DAG), which homed to the transgenic mouse brain at all ages. The authors claim that DAG targets activated astrocytes and that the major DAG receptor is connective tissue growth factor (CTGF). In addition, the authors found that DAG homed to cells in other diseases with acute or chronic neuroinflammation, leading to the conclusion that the DAG peptide is a tool "to improve imaging and management of neuroinflammatory conditions". As the authors state, new approaches are needed towards development of treatments and markers for Alzheimer's disease and other diseases with a neuroinflammatory component, and this kind of study could potentially open such doors. However, some of the major claims of this paper are not fully formed and require additional evidence to support them.

Major issues:

- 1) The authors show results indicating that DAG colocalizes with astrocytes. Although, the authors state that this evidence indicates that DAG is localizing specifically to astrocytes, they also find extracellular DAG and binding to Abeta-positive blood vessels in Tg2576 mice, which seem to indicate that DAG is binding to Abeta as well. Can the authors further discuss the significance of this? In addition, it is unclear from this evidence how the authors think that DAG homes to the brains of these transgenic models as well as those of other diseases with neuroinflammation.
- 2) The authors mention that DAG signal was found in Tg2576 brain in both hippocampus and cortex and that there is DAG homing in young hAPP-J20 animals to cerebral cortex. Does this peptide home to cortex in aged hAPP-J20 mice as well? There is no evidence in the paper of this, which leads the reader to suspect that it has been intentionally left out.
- 3) The authors claim that DAG specifically targets astrocytes. However, they do not offer evidence of what other cell types were examined for DAG. What other cell-specific markers were used to make the determination that DAG is astrocyte-specific?
- 4) In identifying the receptor for DAG, the authors indicate that a number of hits were found through mass spectrometry but they only examine CTGF further. What are the other hits? Why are these less relevant? It's unclear how they conclude that this is *the* DAG receptor in the brain when they have not looked at the other hits. Perhaps there is more than one receptor? Without this additional info and experiments, CTGF is only a candidate receptor and should be classified as such.
- 5) The authors state (page 10; line 233) that: "Importantly, vascular homing appears to be independent of BBB status...homing to the neurovascular unit in hAPP-J20 mice at an early stage of the disease, when the BBB is presumably still intact, supports that conclusion." Where is the evidence that the BBB is still intact in these animals at this stage (reference and/or data)? It is not enough to say "presumably". The authors need data to support this claim.

Minor issues:

- 1) In the introduction, the authors should mention the recent paper (Sevigny et al., 2016 Nature 537, 50–56) on aducanumab, which had positive results as compared with previous Abeta immunotherapies.

REVIEWERS' COMMENTS:

Reviewer #1 (Remarks to the Author):

In the revised manuscript, the authors have addressed the reviewer's comments by discussing or providing new experimental data. The reviewer has no further comments.

Reviewer #3 (Remarks to the Author):

The authors have done a good job of addressing my prior critiques. I have two overall comments; both minor.

1) The legend in Figure 4a is confusing and does not make sense.

2) In Figure 5, with the additional data the authors added, the figure is now somewhat misleading. It would be helpful to have an additional DAG stain with GFAP or CD31 to show cellular orientation to more definitively state that this phenomenon occurs in humans and mice.

Authors' response to reviewers' comments:

All reviewers:

Two reviewers brought up the issue of CTGF vs other hits from the affinity chromatography experiment.

The authors mentioned that CTGF was one of several candidate proteins identified from DAG affinity chromatography. How about other candidates? Are they all located in vascular units? What is the specificity of DAG to vascular units? (Reviewer #1, point 3).

*In identifying the receptor for DAG, the authors indicate that a number of hits were found through mass spectrometry but they only examine CTGF further. What are the other hits? Why are these less relevant? It's unclear how they conclude that this is *the* DAG receptor in the brain when they have not looked at the other hits. Perhaps there is more than one receptor? Without this additional info and experiments, CTGF is only a candidate receptor and should be classified as such. (Reviewer #3, point 4).*

Response: We have now included the other hits from the affinity chromatography in Table 1. We elected to focus initially on CTGF because of the following considerations: 1. Many of the candidate receptors were cytoplasmic proteins or had no reported or plausible connection to AD. 2. CTGF is a heparin-binding matricellular protein, which would make it available for peptide binding in brain extracellular matrix and cell surfaces. 3. Elevated expression of CTGF had been noted in small blood vessels of AD patient brains (*Neuroscience* 2003;116: 1). In addition, elevated CTGF expression takes place in various contexts of brain damage (*Acta Neuropathol* 2003;106: 449; *J Neurosurg Spine* 2005; 2: 319; *Neurotrauma* 2001;18: 377). This logic of focusing on CTGF is now given in the manuscript (p. 7).

Individual reviewers:

Reviewer #1:

1. *How do the authors quantitate the affinity of FAM-DAG to vascular units (such as in endothelial cells or astrocytes)? In Fig 1d, the FAM-DAG scatteredly appears in cells positive for CD31 in hippocampus of hAPP J20 mice (1 out of 7 CD31+ cells co-localizes with FAM-DAG). In Fig 1e, Fig 2 and S3, however FAM-DAG accumulates in endothelial or astrocytes at different extent.*

Response: The reviewer is right in pointing out that the degree DAG accumulates in brain endothelial cells and astrocytes is different in the two AD models. However, in each model, both cell types are positive for DAG accumulation. It is difficult to suggest reasons for the difference because the two models differ from one another in a number of ways, including the APP mutation, the promoter driving the APP production, and the age of onset of the disease. We have now pointed this out in the manuscript (p. 11).

2. *In Fig 1d and e, and Fig S3, why FAM-DAG preferentially locates in hippocampus of hAPP-J20 while it appears to wildly locate in Tg2576 mice. How do the authors explain this brain region preference of DAG peptide? In Fig S3, the FAM-DAG peptide clumps in Tg2576 mouse brains even at the place of which Ab plaques are absent, and reactive astrocytes only locate at the border zones of these clumps. Does it mean that FAM-DAG in fact has other targets or aggregates?*

Response: The first part of the comment is answered under point 1 above. It is true that in Fig S3,

DAG peptide localizes in regions with no visible Abeta plaques, however adjoining GFAP+ astrocytes always appear in these regions. CTGF, the putative receptor for DAG peptide, is secreted by reactive astrocytes as shown by (Ueberham et. al, Neuroscience 2003), and it seems that DAG binds to secreted CTGF that is deposited in the brain ECM. This CTGF deposition and DAG homing were not observed in WT mice. We have now clarified this point in the manuscript (p. 8, line 4).

3. *The authors mentioned that CTGF was one of several candidate proteins identified from DAG affinity chromatography. How about other candidates? Are they all located in vascular units? What is the specificity of DAG to vascular units?*

Response: See response to all reviewers.

4. *In Fig 3, a negative control is needed before the authors conclude that DAG is specific to CTGF and that CTGF is a DAG receptor. Does DAG bind to other members of CCN family proteins or other proteins in astrocytes?*

Response: A control peptide has been added in Figure 3b. It shows no binding to CTGF. Moreover, unlabeled DAG inhibited the binding of labeled DAG to CTGF. This control is generally thought to be the most rigorous test of specificity.

5. *Authors need to provide data confirming the cells they differentiated from hiPSCs are the BMECs. Also, data are needed to support their claims that DAG-AgNPs bind to the BMECs from hiPSCs. From the Fig S10, one cannot judge which cell types the DAG-AgNPs bind to, and the cells are not labeled by any markers.*

Response: Data characterizing the cells as brain endothelial cells has been added (Fig. S13a). The final cell population that we used for our experimentation was over 90 % of BMEC identity; therefore we did not think it was critical to stain the cells with any markers during our AgNP binding experiment.

6. *In Fig 5B, the images of CTGF immunostaining are not consistent with the histogram; the imaging does not reflect about 7-fold increase in the CTGF expression as shown in the histogram.*

Response: We have now reduced the temperature (color tone) of both the images (equally) to 5300K to make the difference between the blue and brown color more clear and it now reflects the 7-fold difference.

7. *Authors described that DAG binds to GFAP+, CD+ cells which are positive for CTGF (Fig. 4B) at the early-stage of AD (3-4 months of age), the time point which Abeta deposition is absent (page 8). However, in human AD (Fig 5C) and mouse AD (Figs. 2 and S3), the authors consistently show that DAG is co-localized with Abeta plaques. Does DAG itself have affinity with Abeta polypeptides? Or, the colocalization between DAG and Abeta plaques is due to the affinity of DAG with reactive astrocytes surrounding the Abeta plaques. This needs to be clarified, as different molecular action of DAG might be involved.*

Response: DAG does not co-localize with the Abeta plaques, rather, DAG accumulates in astrocytes that surround the plaques (note the lack of DAG signal at the center of the Abeta plaque

in Fig 2, panel a). Our data on DAG homing to GFAP+ cells in the early stage J20 animals (when Abeta deposition is absent) agrees with the literature that astrogliosis occurs before Abeta plaque deposition (Wright et al. PLoS One 2013) in the J20 mouse model. Thus, DAG accumulation in AD brain is not due to binding to Abeta. That this is the case is further confirmed by our data on DAG homing in other models of astrogliosis such as glioblastoma and Parkinson's disease. We have now clarified in the manuscript (p.6 and p.13).

8. *Though reactive astrocytes suggest neuroinflammation, the data shown by the authors mainly indicate that the DAG peptide binds to the components of the neurovascular unit, likely reflecting vascular impairment or alteration during AD. Maybe, it is a similar case to PD (the image of PD mouse brains in Fig 6 looks like that FAM-DAG labels vessels). Thus, the authors need more evidence to validate DAG-neuroinflammation aspect of this work, before bringing DAG to a broad indication.*

Response: DAG also accumulates in astrocytes that are not associated with blood vessels (e.g. Fig. 2a), but that express high levels of GFAP, and therefore appear to be reactive astrocytes. However, based on the reviewer's comment, we have toned down the neuroinflammation aspect in the discussion and removed 'neuroinflammation' from the title of the paper.

Minor comments:

9. *The authors i.v. injected a CX7C naïve phage library for 30 min in animal models and then harvest tissues to search peptides. Why do they use 30 min, is there any specific reason or rational for that? There is no explanation on it through the manuscript.*

Response: The rationale for the 30 min is explained in the reference we cite (Mapping of vascular ZIP codes by phage display. Teesalu T, Sugahara KN, Ruoslahti E. Methods Enzymol. 2012;503:35-56). The half-life the phage in the circulation is short (5 min), and there is essentially no phage left in the circulation 30 min after the injection. We have added this description in the methods section of the manuscript (p.16).

10. *Page 5 line 9, the authors did not indicate the figure number regarding the results they described.*

Response: The missing figure number has been added on p. 5.

Reviewer #2:

Major comment: It would greatly improve the manuscript if the authors could show that DAG can be used to deliver any attached molecule of their choosing (drug, radiographic label etc.) in any of the models of neuroinflammation used in the manuscript, in order to demonstrate the usefulness of the peptide.

Response: Our data on delivery of fluorophore labeled DAG and DAG phage particles (in the phage screening) to AD brain partly addresses the question raised here. We have also performed additional experiments to show application of DAG mediated targeting of payload in neurological conditions such as Alzheimer's disease. Our results show increased accumulation of DAG-conjugated iron oxide nanoparticles in the hippocampus and the cortex of hPP-J20 mouse brains (Fig. S14). These nanoparticles represent a prototypical imaging contrast agent, demonstrating the

potential utility of DAG for imaging applications in AD.

Minor comment #1: the authors liberally use the words Parkinson disease, TBI, Glioblastoma while referring to far from perfect mouse models of these disease giving the impression that the peptide homed on cells in the actual diseases (not in the models). This should be made clear especially in the abstract where the authors say: "DAG also homed to cells in human Glioblastoma, Traumatic brain injury and Parkinson's disease brain". The text should read "DAG also homed to cells in mouse models of glioblastoma.....".

Response: We have corrected as suggested (abstract and p.8 and fig. 6).

Minor comment #2: It is not clear how the findings of the manuscript have implications for pathogenesis of Alzheimer's disease and other neurodegenerative diseases. This should be further elaborated upon in the text -or the sentence removed from the abstract.

Response: Alzheimer's disease has a strong vascular component. Studies such as ours that probe the molecular specialization of the neurovascular unit in AD can identify peptides specific to AD. Identifying the receptors for these peptides and studying their function can aid in understanding the complex pathogenesis of AD. For example, we believe that the upregulation of CTGF expression in activated (reactive) astrocytes early in the Alzheimer's disease process and the association of these cells with the vasculature are potentially important findings. Further studies on the function of CTGF in AD are needed to ascertain if CTGF may represent an attractive therapeutic target. These points are now made in the discussion (p.14).

Reviewer #3:

Major issues:

1) *The authors show results indicating that DAG colocalizes with astrocytes. Although, the authors state that this evidence indicates that DAG is localizing specifically to astrocytes, they also find extracellular DAG and binding to Abeta-positive blood vessels in Tg2576 mice, which seem to indicate that DAG is binding to Abeta as well. Can the authors further discuss the significance of this? In addition, it is unclear from this evidence how the authors think that DAG homes to the brains of these transgenic models as well as those of other diseases with neuroinflammation.*

Response: DAG homing to the extracellular regions and endothelial cells in the AD brain colocalizes with CTGF expression (Fig. 3a and S11). CTGF has been reported to be present on endothelial cells (Spilet et al. *Acta Neuropathol* 2003;106: 449;) and also secreted by reactive astrocytes (Ueberham et. al, *Neuroscience* 2003). A majority of reactive astrocytes are found surrounding Abeta deposits, and that makes the DAG localization seem to coincide with Abeta in late stage AD. However, DAG accumulates in the astrocytes that surround the Abeta deposits, not over the actual deposits (note the lack of DAG signal at the center of the Abeta plaque in Fig 2, panel a). Moreover, data in models with no Abeta deposition such as early stage J20 animals or other models of neuroinflammation also agree with the conclusion that DAG does not bind Abeta. This has now been clarified in the discussion (p.13).

2) *The authors mention that DAG signal was found in Tg2576 brain in both hippocampus and cortex and that there is DAG homing in young hAPP-J20 animals to cerebral cortex. Does this peptide home to cortex in aged hAPP-J20 mice as well? There is no evidence in the paper of this,*

which leads the reader to suspect that it has been intentionally left out.

Response: Yes, DAG homes to the cortex of hAPP-J20 mice brains. The missing information has now been added (Fig. S2).

3) *The authors claim that DAG specifically targets astrocytes. However, they do not offer evidence of what other cell types were examined for DAG. What other cell-specific markers were used to make the determination that DAG is astrocyte-specific?*

Response: In the way of clarification, we also showed that DAG also targets endothelial cells. No co-localization of DAG with neurons, or microglia is seen. The data has been added as Fig. S4 and described in the results (p. 6).

4. *In identifying the receptor for DAG, the authors indicate that a number of hits were found through mass spectrometry but they only examine CTGF further. What are the other hits? Why are these less relevant? It's unclear how they conclude that this is *the* DAG receptor in the brain when they have not looked at the other hits. Perhaps there is more than one receptor? Without this additional info and experiments, CTGF is only a candidate receptor and should be classified as such.*

Response: See response to all reviewers on page 1.

5) The authors state (page 10; line 233) that: "Importantly, vascular homing appears to be independent of BBB status...homing to the neurovascular unit in hAPP-J20 mice at an early stage of the disease, when the BBB is presumably still intact, supports that conclusion." Where is the evidence that the BBB is still intact in these animals at this stage (reference and/or data)? It is not enough to say "presumably". The authors need data to support this claim.

Response: We carried out staining of mouse IgG as a marker for BBB leakage in J20 mice. We observed a slight increase that was not significant in 4-month-old hAPP-J20 mice compared to WT (Fig. S12). We have also modified the above-mentioned statement in the manuscript (p.12).

Minor issues:

1) *In the introduction, the authors should mention the recent paper (Sevigny et al., 2016 Nature 537, 50–56) on aducanumab, which had positive results as compared with previous Abeta immunotherapies.*

Response: The citation has been added in the introduction (p.3).

REVIEWERS' COMMENTS:

Reviewer #1 (Remarks to the Author):

In the revised manuscript, the authors have addressed the reviewer's comments by discussing or providing new experimental data. The reviewer has no further comments.

Reviewer #3 (Remarks to the Author):

The authors have done a good job of addressing my prior critiques. I have two overall comments; both minor.

1) The legend in Figure 4a is confusing and does not make sense.

Response: The legend in fig 4a has been modified to make it clearer

2) In Figure 5, with the additional data the authors added, the figure is now somewhat misleading. It would be helpful to have an additional DAG stain with GFAP or CD31 to show cellular orientation to more definitively state that this phenomenon occurs in humans and mice.

Response: We have added an additional panel in Figure 5 that shows DAG binding co-localizes with GFAP staining in human brain tissue.